# DIMAFv2: Interpretable Multimodal Disentangled Representations for Breast Cancer Survival Prediction

Aniek Eijpe[*1], Soufyan Lakbir[1], Melis Erdal Cesur[2], Sara P. Oliveira[2], Sanne Abeln[1], and Wilson Silva[1]

[1]AI Technology for Life, Department of Information and Computing Sciences, Department of Biology, Utrecht University, Utrecht, The Netherlands
a.eijpe@uu.nl
[2]Computational Pathology group, Department of Pathology, The Netherlands Cancer Institute, Amsterdam, The Netherlands

## Abstract

Understanding tumor biology and predicting patient survival is challenging due to the complex and heterogeneous information across multiple modalities. We propose an interpretable framework that integrates histopathology whole-slide images and transcriptomics data by explicitly separating modality-shared and modality-specific representations through disentangled attention fusion. Our approach achieves state-of-the-art breast cancer survival prediction and enhanced disentanglement. DIMAFv2's inherent interpretability revealed important factors driving prediction, such as KRAS signaling and G2M checkpoint pathways interacting with tumor morphology.

## 1 Introduction

Effective cancer survival prediction remains challenging due to the complex, heterogeneous nature of tumor biology [1]. While unimodal models using histopathology Whole Slide Images (WSI) or transcriptomics data have achieved strong results [2–4], multimodal approaches can capture richer biological context and improve prediction performance [5–8].

However, most multimodal approaches focus on the shared information across modalities, potentially suppressing important complementary information [8, 9]. Disentangled Representation Learning (DRL) [10, 11] offers a way to explicitly separate modality-specific and modality-shared information in disentangled representations, also enhancing interpretability. Yet, current DRL-based models rely on linear or network-based disentanglement objectives [8, 12, 13] and rarely quantify or analyze the disentanglement and interpretability achieved.

To address this, we previously proposed **Disentangled and Intepretable Multimodal Attention Fusion (DIMAF)** [9], a framework that separates intra- and inter-modal interactions between WSIs and transcriptomics and promotes disentanglement between the resulting modality-specific and modality-shared representations via a Distance Correlation (DC) loss. Extending this, we introduce **DIMAFv2**, which uses refined WSI representations and an attention-based aggregation within the disentangled representations to improve interpretability and disentanglement. Moreover, we perform an interpretability analysis to study how modality-shared and modality-specific features emerge from unimodal interactions and drive survival outcomes.

## 2 Methods

We developed an interpretable deep learning model that predicts survival risk by integrating H&E-stained WSIs and bulk RNA-Sequencing (RNA-Seq) data. Following our prior work [9], we create 2 unimodal representations, one containing 50 RNA-seq features and the other 16 WSI features. The RNA-seq features correspond to known pathways with well-defined biological functions [14]. The WSI features represent distinct morphological prototypes, which were annotated by a pathologist, and their cardinality [4, 7]. In DIMAFv2, we modify the WSI encoding to better capture prototype cardinality by adding this directly before multimodal fusion.

The unimodal features are fused in the Disentangled Attention Fusion layer, which models the intra-modal interactions using two self-attention layers and the inter-modal interactions using two cross-attention layers, creating two modality-specific and two modality-shared representations, respectively [9]. In other words, each unimodal feature generates two multimodal features: one enriched with the other modality (modality-shared) and one with the same modality (modality-specific). The multimodal features in the obtained representations are pooled via an attention-based aggregation layer [15], forming the final four disentangled representations. Using these representations and a linear predictor, we obtain a risk score $r$. The overall loss objective is

$$\mathcal{L} = \mathcal{L}_{surv} + \lambda_{dis} \cdot \mathcal{L}_{dis} \quad (1)$$

where $\mathcal{L}_{surv}$ is the Cox partial log likelihood loss [16, 17], $\lambda_{dis} = 7$ and $\mathcal{L}_{dis}$ the DC-based [18, 19] loss promoting disentanglement between the modality-specific and shared representations [9].

---

[*]Corresponding Author.

**Table 1.** DSS test results on TCGA-BRCA with **best** and second-best performances. The p-value corresponds to the log-rank test between the predicted risk cohorts.

|  | C-index | C-index-IPCW | p-value |
|---|---|---|---|
| PIBD [8] | $0.608 \pm 0.100$ | $0.497 \pm 0.137$ | $7.95e^{-2}$ |
| MMP [7] | $0.741 \pm 0.056$ | $0.519 \pm 0.185$ | $9.04e^{-5}$ |
| DIMAF [9] | $0.759 \pm 0.065$ | **$0.580 \pm 0.161$** | $4.08e^{-5}$ |
| DIMAFv2 | **$0.760 \pm 0.057$** | $0.553 \pm 0.150$ | **$2.80e^{-5}$** |

**Table 2.** Disentanglement test results on TCGA-BRCA with **best** and second-best performances.

|  | Distance Correlation | Orthogonal Score |
|---|---|---|
| PIBD [8] | $0.624 \pm 0.021$ | $0.246 \pm 0.034$ |
| DIMAF [9] | $0.356 \pm 0.045$ | $0.066 \pm 0.009$ |
| DIMAFv2 | **$0.333 \pm 0.041$** | **$0.060 \pm 0.016$** |

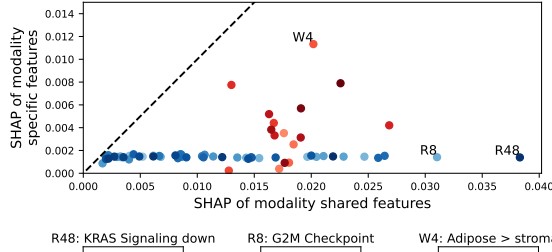

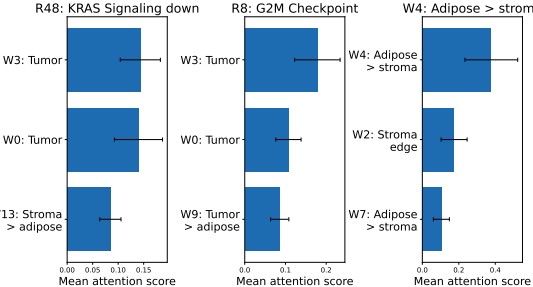

**Figure 1.** Interpretability analysis of DIMAFv2. **Top:** Shows the mean absolute SHAP for the modality-shared versus modality-specific features derived from each pathway feature (blue) and morphological prototype feature (red). Top features in either category are annotated. **Bottom:** Top mean attention weights contributing to the highlighted features, with error bars representing the standard deviation across test samples.

We trained the model to predict disease-specific survival (DSS) risk using 5-fold cross-validation on the TCGA-BRCA (Breast Invasive Carcinoma) dataset for 30 epochs using the AdamW optimizer.

We evaluated the model's survival prediction performance using the C-index, its weighted variant C-index-IPCW [20], and performed the log-rank test between high and low risk cohorts. Moreover, we assessed total representation disentanglement using DC and an orthogonal score, defined as the absolute cosine similarity. Lastly, we applied DeepSHAP [21] to quantify the overall contribution of modality-shared and modality-specific features, and we visualized the attention weights to reveal the inter- and intra-modal interactions driving survival prediction.

## 3 Results

Table 1 shows that our proposed framework, DIMAFv2, achieves the highest average c-index and the lowest log-rank p-value. For the C-index-IPCW, DIMAFv2 ranks second, behind DIMAF [9].

Table 2 shows that DIMAFv2 further improves disentanglement between modality-specific and modality-shared representations, achieving a lower DC and orthogonal score than DIMAF. Both substantially improve over PIBD, demonstrating the effectiveness of our disentanglement strategy.

Figure 1 shows how modality-shared and modality-specific features contribute to breast cancer survival prediction. For each unimodal feature (e.g., W4, the WSI prototype annotated as adipose tissue with stroma), we plot the SHAP values of its corresponding modality-shared against its modality-specific feature. Overall, modality-shared features play a more dominant role in driving survival predictions, consistent with previous observations [9]. In particular, the modality-shared KRAS signaling down and G2M checkpoint pathway features are highly predictive, primarily through interactions with tumor morphological prototypes. The top modality-specific features are non-tumor prototypes, such as W4, arising from interactions with non-tumor prototypes.

## 4 Discussion

In this work, we introduced DIMAFv2, which integrates WSIs and transcriptomics data through disentangled attention fusion. We demonstrated improved survival prediction performance while enhancing disentanglement between modality-shared and modality-specific representations. The interpretable nature of DIMAFv2 allows us to identify which multimodal features (modality-specific or modality-shared) and interactions drive breast cancer survival prediction. We found that the modality-shared KRAS signaling and G2M checkpoint features, primarily enriched via interactions with tumor prototypes, are important for survival prediction in DIMAFv2. Both pathways are associated with breast cancer subtypes and survival outcomes [22, 23]. Moreover, we found that the modality-specific prototype of adipose tissue with stroma was also important, which is consistent with previous studies suggesting that breast adipose tissue plays a major role in breast cancer risk [24]. Overall, these findings demonstrate that DIMAFv2 can both predict clinical outcomes and provide meaningful insights into multimodal cancer biology.

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
