# OpenReview forum: "DIMAFv2: Interpretable Multimodal Disentangled Representations for Breast Cancer Survival Prediction"
_NLDL.org/2026/Abstracts_Track — NLDL 2026 Abstracts_

### Official Review · Reviewer_2Mbn · 2025-10-31

**Soundness:** 4
**Correctness:** 4
**Rating:** 5
**Confidence:** 4

**Summary:**

The paper presents DIMAFv2, an improved version of a prior model aimed at multimodal representation learning. The key contributions are: (i) enhanced interpretability that is shown to align with established biological knowledge, and (ii) improved modality disentanglement, reportedly with minimal performance regressions and, in some cases, performance gains over previous approaches.

**Strengths:**

- Comprehensive evaluation protocol: The authors demonstrate that DIMAFv2 outperforms competitive baselines, suggesting that the proposed architectural and training changes are effective.
- Interpretability grounded in biology: The interpretability analyses are shown to be consistent with known biological mechanisms, lending credibility to the explanations and suggesting the model is capturing meaningful structure rather than spurious correlations.

**Weaknesses:**

- Missing SOTA comparison on disentanglement: While the paper emphasizes improved modality disentanglement, it does not compare disentanglement quality against strong contemporary baselines.
- Single-dataset evaluation: Results are reported for only one dataset. This limit claims about generalization, robustness, and the reliability of both the performance and interpretability findings across domains, cohorts, or acquisition protocols.
- Reproducibility gaps (no code or split information): The absence of code and dataset split details hampers reproducibility and may inadvertently inflate performance if, for example, there is unintentional leakage or non-standard preprocessing. Precise split definitions (e.g., patient-level splits, cross-validation folds), preprocessing pipelines, and random seeds are needed.
- Unclear treatment of modality-shared representations: The manuscript states there are four different representations but does not specify how the modality-shared component differs when derived from each input modality.

- Minor presentation issues:
  - Make the x-axis scale consistent across panels in Figure 1.
  - Add a legend to Figure 1 for clarity.

---

### Official Review · Reviewer_NDCt · 2025-11-02

**Soundness:** 3
**Correctness:** 3
**Rating:** 4
**Confidence:** 3

**Summary:**

The paper proposes an interpretable framework that integrates histopathology whole-slide images and transcriptomics data to perform breast cancer survival prediction. The approach uses disentangled representation learning to separate modality-specific and modality-shared information. It extends on the authors’ previous work (DIMAF), where the approach presented here (DIMAFv2) refines whole-slide image representations and uses an attention-based aggregation to improve interpretability and disentanglement. Results show some improvement in prediction, while allowing for identification of the multimodal features contributing to the prediction.

**Strengths:**

The ideas are interesting and show potential for improved interpretability and disentanglement.

**Weaknesses:**

The abstract compresses very much information into limited space and could have benefitted from some more focus. The values for the C-index-IPCW actually seem quite low, and the DIMAFv2 does not seem significantly better than DIMAF when it comes to the actual prediction. Figure 1, top, could be better explained.

---

### Official Review · Reviewer_vXLf · 2025-11-03

**Soundness:** 3
**Correctness:** 2
**Rating:** 4
**Confidence:** 3

**Summary:**

The abstract presents DIMAFv2, an extension to the *Disentangled and Intepretable Multimodal Attention Fusion* (DIMAF) framework for breast cancer survival prediction. The extension is designed to enhance the framework's retention of complementary information from multiple modalities, by introducing an attention-based aggregation mechanism. The abstract claims to achieve state-of-the-art breast cancer survival prediction performance, as well as inherently better interpretability than the baseline.

**Strengths:**

1. __Writing and presentation.__ The abstract is well written and easy to follow.
2. __Clarity.__ The authors present a logical extension to the DIMAF baseline. The introduced attention-based enhancement of the disentanglement mechanism is clearly motivated.
3. __Interpretability.__ DIMAFv2 focuses on interpretability, which is crucial for informing biomedical professionals.

**Weaknesses:**

1. __Marginal performance gain.__ The reported improvement over DIMAF is negligible (ΔC-index ≈ 0.001) and likely within cross-validation variance.
No statistical significance test is provided. This weakens claims of meaningful performance improvement.
2. __Limited scientific novelty.__ DIMAFv2 mainly introduces incremental refinements (refined WSI encoding and attention pooling) rather than a new conceptual framework.
The contribution feels methodological rather than scientific.
3. __No external validation.__ All results are limited to TCGA-BRCA.
There is no evidence of generalization to independent datasets (e.g., METABRIC, CPTAC) or to other cancer types.
4. __Interpretability not human-centered.__ The interpretability analysis is confined to SHAP values and attention maps.
These are machine-centric measures and do not demonstrate whether explanations are understandable or useful to pathologists or biologists.
5. __Post hoc biological interpretation.__ The discussion primarily confirms known pathways (KRAS, G2M) rather than revealing novel biological insights.
Thus, interpretability serves as confirmation rather than discovery.
6. __Lack of reflection on limitations.__ The abstract only highlights positive results.
No discussion of potential biases (e.g., single pathologist prototype annotation), dataset limitations, or clinical applicability.
7. __Technical jargon.__ The abstract is overloaded with technical jargon (“disentangled attention fusion”, “distance correlation loss”) and may be difficult for biomedical audiences to follow.

---

### Decision · Program_Chairs · 2025-11-05

**Decision:**

Accept

**Comment:**

The abstract is of interest to the community and should be presented at the conference.